# ATP-dependent hydroxylation of an unactivated primary carbon with water

Christian Jacoby [1], Sascha Ferlaino [2], Dominik Bezold [3], Henning Jessen [3], Michael Müller [2] & Matthias Boll [1✉]

Enzymatic hydroxylation of unactivated primary carbons is generally associated with the use of molecular oxygen as co-substrate for monooxygenases. However, in anaerobic cholesterol-degrading bacteria such as *Sterolibacterium denitrificans* the primary carbon of the isoprenoid side chain is oxidised to a carboxylate in the absence of oxygen. Here, we identify an enzymatic reaction sequence comprising two molybdenum-dependent hydroxylases and one ATP-dependent dehydratase that accomplish the hydroxylation of unactivated primary C26 methyl group of cholesterol with water: (i) hydroxylation of C25 to a tertiary alcohol, (ii) ATP-dependent dehydration to an alkene via a phosphorylated intermediate, (iii) hydroxylation of C26 to an allylic alcohol that is subsequently oxidised to the carboxylate. The three-step enzymatic reaction cascade divides the high activation energy barrier of primary C–H bond cleavage into three biologically feasible steps. This finding expands our knowledge of biological C–H activations beyond canonical oxygenase-dependent reactions.

[1] Microbiology, Faculty of Biology, Albert-Ludwigs-Universität Freiburg, Schänzlestr. 1, 79104 Freiburg, Germany. [2] Institute of Pharmaceutical Sciences, Albert-Ludwigs-Universität Freiburg, Albertstrasse 25, 79104 Freiburg, Germany. [3] Institute of Organic Chemistry, Albert-Ludwigs-Universität Freiburg, Albertstrasse 21, 79104 Freiburg, Germany. ✉email: matthias.boll@biologie.uni-freiburg.de

The selective oxidation of unactivated C–H bonds at alkyl functionalities to alcohols is of great importance for a plethora of synthetic processes. Enzymatic solutions for these challenging reactions have continuously motivated organic chemists for developing bioinspired strategies[1–5]. Biocatalytic C–H bond activations of alkyls via hydroxylation have generally been associated with metal-dependent monooxygenases, peroxidases or peroxygenases[6–8]. As an example, the well-studied cytochrome P450 monooxygenases reduce the dioxygen co-substrate to the formal oxidation state of a hydroxyl radical, which allows for the hydroxylation of unactivated primary carbons to alcohols[9]. While cytochrome P450 enzymes require an auxiliary electron donor system, peroxygenases and peroxidases depend on a balanced supply with its reactive co-substrate[6,8].

The use of oxygenases or peroxygenases is not an option for C–H bond oxidations in anaerobic hydrocarbon-degrading microorganisms, for which a few oxygen-independent enzymatic solutions for C–H bond hydroxylation at activated positions have been discovered in the past decades[10]. Prototypical examples are the water-dependent hydroxylations of the benzylic carbons of p-cresol[11,12], p-cymene[13], and ethylbenzene[14,15], as well as of the allylic carbon of limonene[16]. Here, C–H bond activation proceeds via hydride transfer to FAD or Mo(VI) = O cofactors (Fig. 1a–d). In all these reactions the resulting carbocation intermediates are stabilised by the possibility of forming multiple resonance structures. However, the water-dependent hydroxylation of unactivated primary carbons of alkanes or isoprenoids has not been reported so far in enzyme catalysis.

The ubiquitous, biologically active steroids are composed of a tetracyclic steran system and an isoprenoid side chain resulting in a low water-solubility and persistence in the environment. Bacterial degradation of steroids is of global importance for biomass decomposition, removal of environmental pollutants and for intracellular survival of Mycobacterium tuberculosis and other

pathogens[17–19]. In aerobic cholesterol degrading bacteria, side chain oxidation is initiated by the hydroxylation of the unactivated primary C26 (or equivalent C27) to a primary alcohol by cytochrome P450 enzymes that is further oxidised to a C26-carboxylate[20,21]. The isoprenoid side chain of the latter is then converted to acetyl-CoA and propionyl-CoA units via modified β-oxidation[22,23]. In denitrifying bacteria, oxygen-independent cholesterol degradation also proceeds via β-oxidation of the 26-carboxylate intermediate involving highly similar enzymes[24–26]. However, the oxidation of the primary, unactivated C26 to the 26-carboxylate must proceed in an oxygen-independent manner.

In the denitrifying β-proteobacterial model organism Sterolibacterium denitrificans Chol-1S, anaerobic cholesterol degradation is initiated by the periplasmic isomerization/dehydrogenation of ring A to cholest-4-en-3-one (CEO) by the AcmA gene product[27], that may be further dehydrogenated to cholesta-1,4-dien-3-one (CDO) by AcmB[25]. Both intermediates serve as substrate for steroid C25 dehydrogenases (S25DH) that hydroxylates tertiary C25 of the isoprenoid side chain with water to 25-OH-CEO/-CDO in the presence of cytochrome c or an artificial electron acceptor[28] (Fig. 1e). Periplasmic S25DH belongs to the DMSO reductase family of Mo-cofactor containing enzymes and is composed of the catalytic α-subunit harbouring the Mo-cofactor and the FeS-clusters/heme b containing electron transferring β- and γ-subunits[29]. This enzyme is proposed to abstract a hydride from tertiary C25 to the Mo(VI) = O species yielding the Mo(IV)–OH intermediate[14]. The latter transfers the hydroxyl group then back to the assumed tertiary carbocation intermediate; electrons are transferred to cytochrome c. The genome of S. denitrificans contains eight homologous genes putatively encoding catalytic α-subunits of S25DH-like enzymes, four of which (S25DH_{1-4}) have been heterologously produced together with the corresponding β- and γ-subunits and an essential chaperone. They specifically hydroxylate side chains of

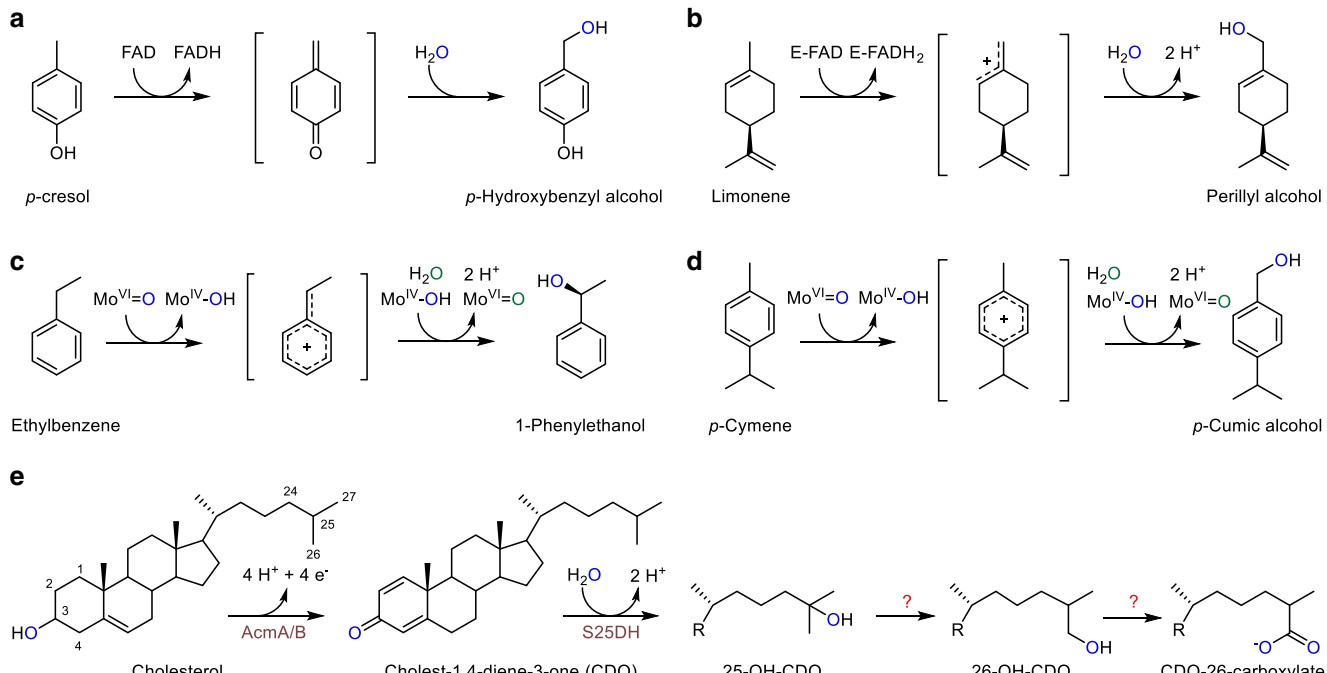

**Fig. 1 Enzymatic water-dependent alkyl hydroxylations. a** p-Cresol methyl hydroxylase. **b** Limonene dehydrogenase. **c** Ethylbenzene dehydrogenase. **d** p-Cymene dehydrogenase. **e** Initial steps of anaerobic cholesterol degradation. The enzyme cascade involved in the proposed conversion of 25-OH-CDO (25-OH-cholesta-1,4-dien-3-one) to CDO-26-carboxylate (cholesta-1,4-dien-3-one-26-carboxylate) via a 26-OH-CDO (26-OH-cholesta-1,4-dien-3-one) intermediate was studied in this work.

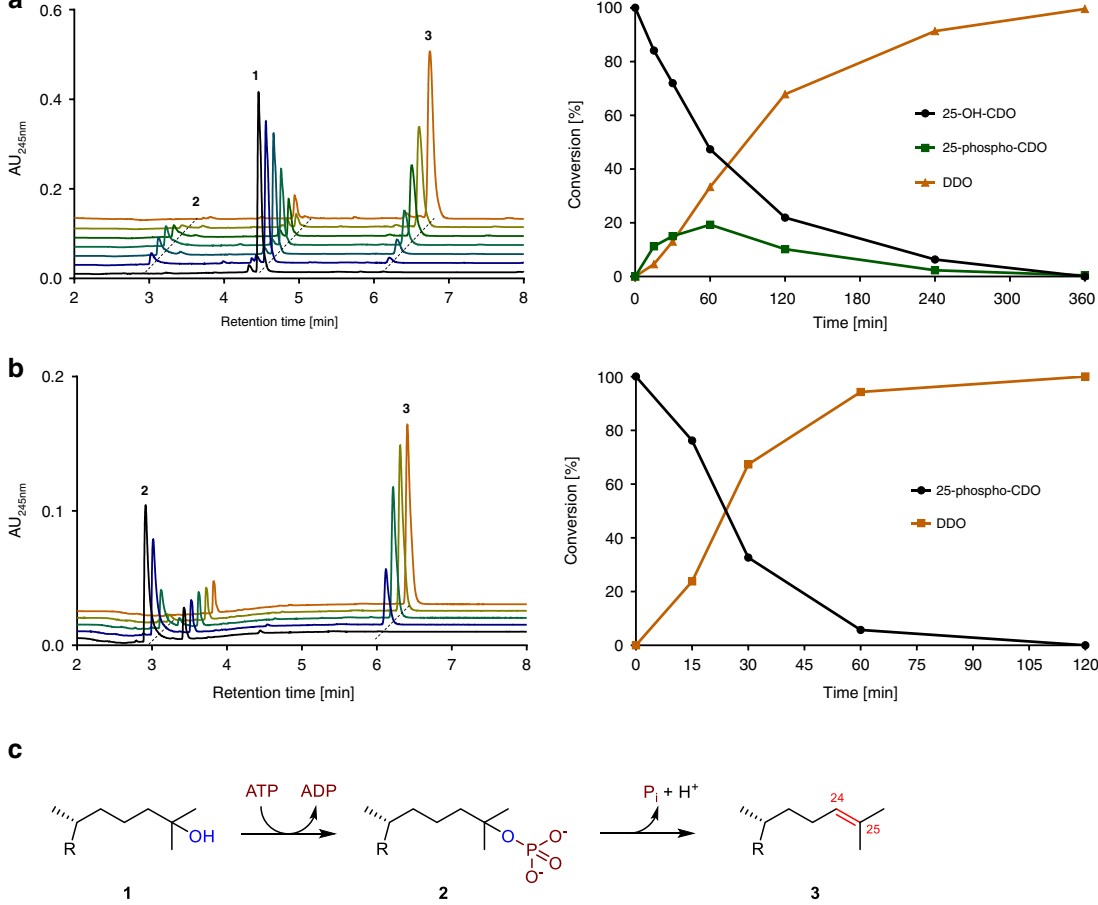

**Fig. 2 Oxygen-independent conversion of 25-OH-cholesta-1,4-dien-3-one (1). a** Time-, ATP-, and MgCl$_2$-dependent conversion of 0.5 mM 25-OH-CDO in soluble cell extracts of *S. denitrificans*. **b** ATP-independent conversion of chemically synthesised 25-phospho-CDO. **c** Proposed reaction sequence of ATP-dependent dehydration of 25-OH-CDO **1** via 25-phospho-CDO **2** to desmost-1,4-diene-3-one (DDO, **3**). Source data are provided as a Source Data file.

cholesterol and its analogues to tertiary alcohols[30,31]. The rationale for the initial formation of the tertiary alcohol 25-OH-CDO, that cannot be directly further oxidised, during anaerobic cholesterol catabolism has remained enigmatic. Though previous high resolution mass spectrometry (HRMS)-based analyses suggested the primary alcohol 26-OH-CDO as an intermediate[24,32], the underlying conversion of a tertiary alcohol to a primary alcohol involves an unknown enzymology.

Here, we aim to identify the enzymatic steps involved in the water-dependent oxidation of a primary carbon atom during anaerobic cholesterol degradation. We demonstrate that the formation of a tertiary alcohol at the cholesterol side chain initiates a three-step enzymatic reaction cascade finally resulting in the oxidation of unactivated C26 to a carboxylate.

## Results

**Enzymatic dehydration of 25-OH-cholesta-1,4-diene-3-one**. To elucidate the enzymology responsible for production of CDO-26-oate, we synthesised 25-OH-CDO from CEO using overproduced S25DH and AcmB. Cell-free extracts of *S. denitrificans* grown with cholesterol under denitrifying condition were anoxically reacted with 25-OH-CDO in the presence of a multitude of natural and artificial electron acceptors (e.g., NAD[P]$^+$, K$_3$[Fe(CN)$_6$, or 2,6-dichlorophenolindophenol at 0.5 mM, respectively). In no case, conversion of 25-OH-CDO to a product was observed.

In further trials, electron acceptors were omitted in the reaction assays and a number of potential co-substrates were tested. In the presence of ATP and MgCl$_2$ (5 mM each), the time-dependent conversion of 25-OH-CDO **1** to a minor, more polar intermediate **2** was observed using ultra-fast performance liquid chromatography (UPLC) analyses, which was readily further converted to a second less polar product **3**, (Fig. 2). After ultracentrifugation of cell-free extracts this conversion was found to occur only in the soluble protein fraction. UPLC- HRMS analysis of **2** did not allow a clear assignment to a molecular mass, whereas analysis of compound 3 revealed a [M + H]$^+$ ion with $m/z$ = 381.3161 ± 0.4 Da suggesting the loss of H$_2$O from the substrate ($m/z$ = 399.3274 ± 1.1; for MS data of all steroid intermediates analysed in this work, see Supplementary Table 1). The product **3** was purified by preparative HPLC and identified by $^1$H, $^{13}$C, and 2D NMR analyses as desmost-1,4-diene-3-one (DDO) containing an allylic Δ24 double bond (Supplementary Figs. 1–3). Thus, the hydroxyl functionality introduced by S25DH was subsequently eliminated to an alkene by an ATP-dependent 25-OH-CDO dehydration activity.

We tested whether product **2** represents a phosphorylated intermediate during the ATP-dependent dehydration of 25-OH-CDO. For this purpose we chemically synthesised a 25-phospho-CDO standard from enzymatically prepared 25-OH-CDO. We opted for a P(III)-amidite approach due to the steric hindrance of the tertiary OH, using a bis fluorenylmethyl (Fm) protected

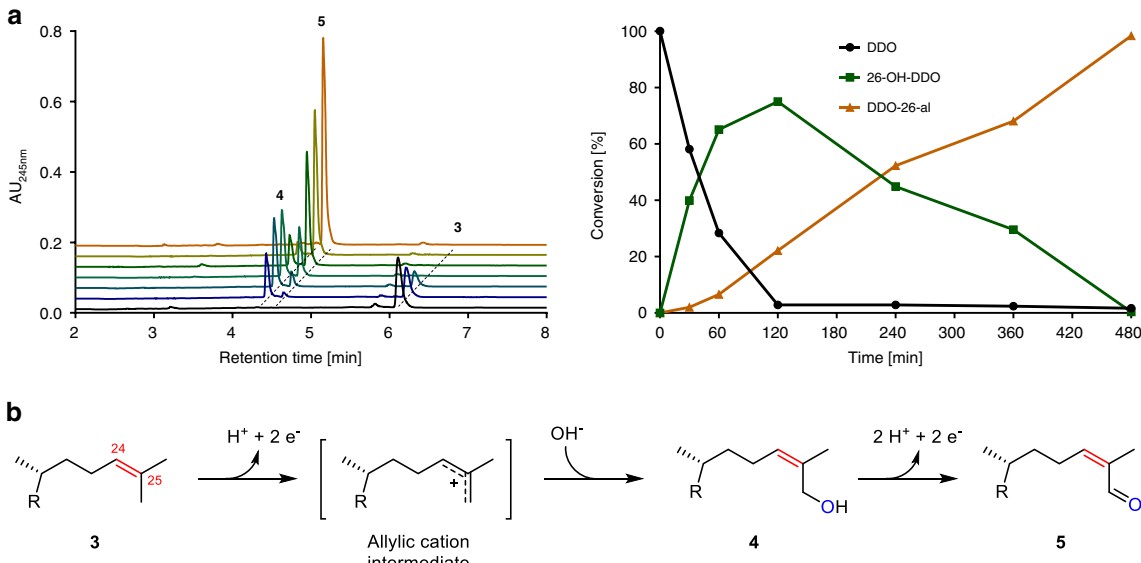

**Fig. 3 Oxygen-independent conversion of desmost-1,4-dien-3-one (3). a** Conversion of DDO to the two products **4** and **5** in soluble extracts of *S. denitrificans*. **b** Reactions sequence obtained via a proposed allylic cation intermediate after NMR-based identification of products as 26-OH-DDO **4** and the corresponding C26-aldehyde DDO-26-al **5**. The enzyme involved was identified as a Mo-dependent hydroxylase. Source data are provided as a Source Data file.

P-amidite[33] that indeed facilitated the reaction. After oxidation and removal of the Fm groups with 1,8-diazabicyclo[5.4.0]undec-7-en (DBU), the product was obtained in a one-flask operation without elimination of the labile phosphate in 39% yield after purification (for MS and NMR analyses, see Supplementary Figs. 4 and 5). When 25-phospho-CDO (**2**) was incubated with soluble *S. denitrificans* extracts, the ATP-independent conversion into DDO (**3**) was observed (Fig. 2b). The kinetic parameters of 25-OH- and 25-phospho-CDO conversions were $V_{max} = 11.7 \pm 0.6$ nmol min$^{-1}$ mg$^{-1}$/$K_m = 41 \pm 8$ µM for 25-OH-CDO, and $V_{max} = 23.2 \pm 1$ nmol min$^{-1}$ mg$^{-1}$/$K_m = 227 \pm 32$ µM for 25-phospho-CDO (mean values ± standard deviations of enzyme activity measurements at various substrate concentrations, for the data points used see Supplementary Fig. 6a, b). In summary, the results indicate that kinase and phosphate elimination partial activities are involved in the conversion of 25-OH-CDO (**1**) to DDO (**3**) via 25-phospho-CDO (**2**) (Fig. 2c).

For the determination of cofactor specificity and stoichiometry, the ATP-dependent 25-OH-CDO dehydration activity was enriched via ammonium sulphate precipitation (activity retained in the suspended 50% saturation pellet). During formation of DDO from 25-OH-CDO no other nucleoside triphosphate than ATP was accepted as co-substrate; Mg$^{2+}$ could substitute for Mn$^{2+}$ albeit at only 5% activity. UPLC analyses of adenosine nucleotides revealed that $1.2 \pm 0.1$ mol ATP were hydrolysed to $0.9 \pm 0.1$ mol ADP and $0.3 \pm 0.1$ mol AMP per mol 25-OH-CDO consumed (mean value ± standard deviation in three independent determinations); ATP hydrolysis was negligible in the absence of the steroid.

**Oxidation of the allylic methyl group**. For analysing the further conversion of DDO, we enzymatically synthesised DDO from 25-OH-CDO using cell-free extracts from *S. denitrificans*. Strictly depending on K$_3$[Fe(CN)$_6$] as electron acceptor, cell-free extracts of *S. denitrificans* converted DDO to product **4** that was subsequently converted to **5** in assays performed anaerobically (Fig. 3).

Compounds **4** and **5** were analysed by UPLC-HRMS and $^1$H, $^{13}$C, and 2D NMR spectroscopy. The products were identified as 26-OH-DDO **4**, and the corresponding aldehyde DDO-26-al **5** (Supplementary Figs. 7–12). Notably, NOE NMR analyses clearly showed that the hydroxyl functionality was stereoselectively introduced in the cis position relative to the C24 proton giving the (E)-C26-OH-DDO and not the (Z)-isomer.

In summary, DDO **3** formed by ATP-dependent dehydration of the tertiary alcohol was hydroxylated first to the allylic primary alcohol **4** with water, followed by the oxidation to the aldehyde **5**. Specific activities in cell-free extracts were $2 \pm 0.1$ nmol min$^{-1}$ mg$^{-1}$ for 26-OH-DDO formation and $1.2 \pm 0.2$ nmol min$^{-1}$ mg$^{-1}$ for 26-OH-DDO dehydrogenation (mean value ± standard deviation in three biological replicates of cell extract preparation).

**Heterologous production of desmost-1,4-dien-3-one (DDO) hydroxylase**. In addition to the previously investigated S25DH$_{1-4}$ isoenzymes, the genome of *S. denitrificans* contains four gene clusters encoding the putative αβγ-subunits of related Mo-dependent dehydrogenases (DH$_{5-8}$)[29,30]. Among these, the α-subunits DH$_{5-7}$ form a phylogenetic subcluster with amino acid sequence identities >80%; that of DH$_8$ is rather distantly related (Fig. 4a).

To test whether one of the DH$_{5-7}$ enzymes may be involved in the water-dependent hydroxylation of DDO, we heterologously expressed the genes potentially encoding the active site αβγ-subunits of DH$_{5-7}$ and an assumed δ-chaperone in the β-proteobacterium *Thauera aromatica* as described for S25DH$_{1-4}$[30]. Using cell-free extracts from *T. aromatica* producing DH$_{5-7}$, only with DH$_5$ we observed the K$_3$[Fe(CN)$_6$] dependent conversion of DDO to 26-OH-DDO; we therefore refer this enzyme to S26DH$_1$. Heterologously produced S26DH$_1$ was enriched from *T. aromatica* extracts by three chromatographic steps under anaerobic conditions using a modified protocol established for S25DH$_{1-4}$[30].

SDS–PAGE analysis of enriched S26DH$_1$ revealed four major protein bands (Fig. 4b) that were excised, tryptically digested, and analysed by electrospray ionisation quadrupole time-of-flight

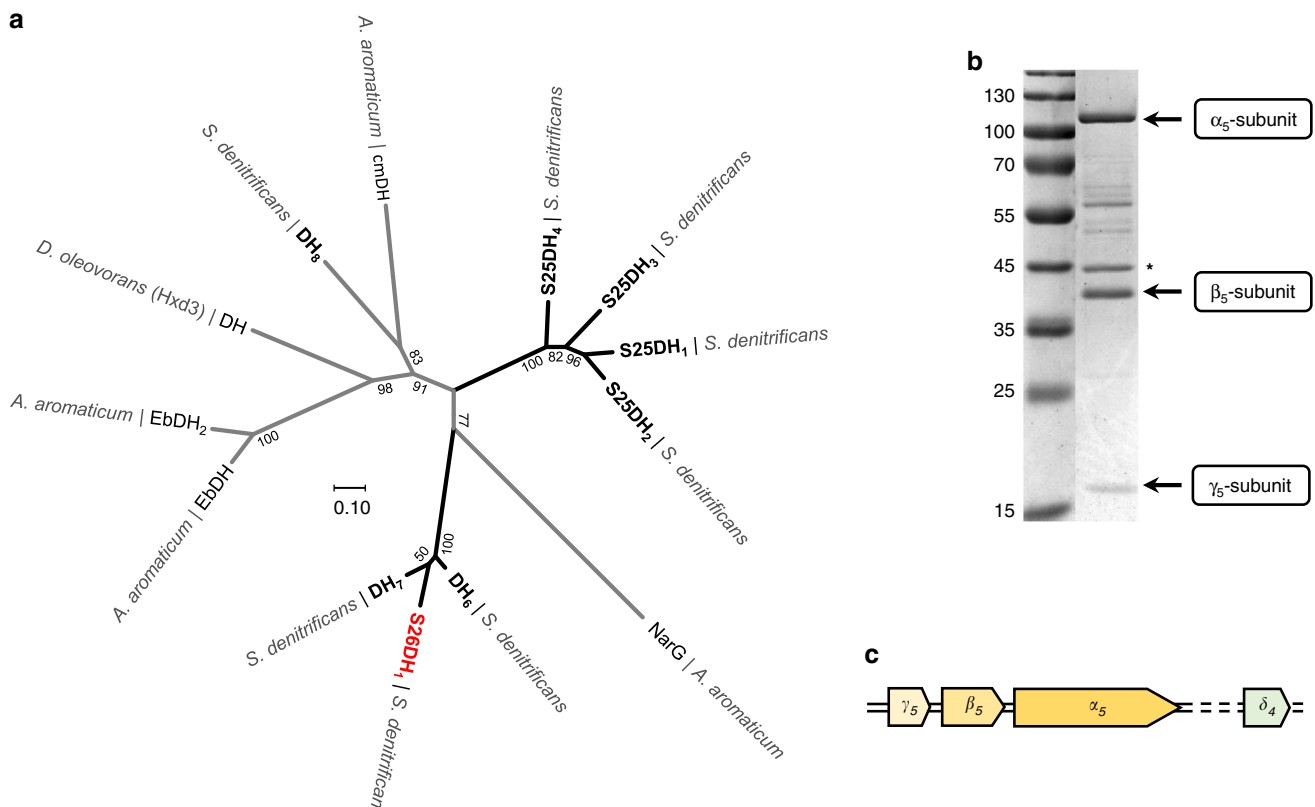

**Fig. 4 Phylogeny and enrichment of recombinant S26DH₁ from *S. denitrificans*. a** Phylogenetic tree of S25DH-like Mo-enzymes of the DMSO reductase family. *A.* = *Aromatoleum*; *D.* = *Desulfococcus*; α-subunits of CmDH = *p*-cymene dehydrogenase; EbDH = ethylbenzene dehydrogenase; NarG = nitrate reductase. The previously assigned DH₅ is now referred to as S26DH₁ (marked in red). **b** SDS–PAGE of S26DH₁ heterologously produced in *T. aromatica*; the numbers refer to the molecular masses (kDa) of a standard. The band marked with an asterisk was identified as co-purified 6-oxocyclohex-1-ene-1-carboxyl-CoA hydrolase from *T. aromatica*; the minor band eluting slightly above the 55 kDa is a degradation product of the α₅-subunit (Supplementary Table 2). **c** Cluster of genes encoding the αβγ-subunits and the gene encoding the δ-chaperone of S26DH₁ (accession numbers: γβα-subunits, WP_154715926-8; δ-chaperone, WP_154716737). Source data are provided as a Source Data file.

mass spectrometry (ESI-Q-TOF-MS). The masses of the ions obtained from tryptic peptides confirmed the presence of the α,β, γ-subunits of S26DH₁ from *S. denitrificans*, those obtained from the band eluting at 43 kDa were assigned to a co-purified 6-oxocyclohex-1-ene-1-carboxyl-CoA hydrolase from *T. aromatica* (Supplementary Table 2). The latter cofactor-free enzyme plays a role in the catabolic benzoyl-CoA degradation pathway[34] and is unlikely to be associated with steroid hydroxylation.

Amino acid sequence similarities of S26DH₁ with characterised S25DHs[30] suggested the presence of one Mo-cofactor, four [4Fe-4S] clusters, one [3Fe-4S] cluster, and a heme *b*. The metal content of S26DH₁ determined by colorimetric procedures was 0.81 ± 0.03 mol Mo, 19.7 ± 0.4 mol Fe, and 0.89 ± 0.01 mol heme *b* per mol S26DH₁, respectively (mean value ± standard deviation in three different enzyme preparations). These values are in good agreement with the expected cofactor content and with the values determined for other S25DHs[30].

Unexpectedly, enriched S26DH₁ catalysed the hydroxylation of DDO to 26-OH-DDO **4** and the subsequent dehydrogenation of the latter to DDO-26-al **5** with $K_3[Fe(CN)_6]$ as electron acceptor (Supplementary Fig. 13). As a control, *T. aromatica* extracts containing the plasmid without the S26DH₁ genes did not catalyse such a reaction. Both partial reactions of S26DH₁ dehydrogenase followed Michaelis–Menten kinetics with $V_{max}$ = 219 ± 11 nmol min⁻¹ mg⁻¹/$K_m$ = 123 ± 25 μM for DDO hydroxylation, and $V_{max}$ = 73 ± 3 nmol min⁻¹ mg⁻¹/$K_m$ = 83 ± 12 μM for 26-OH-DDO dehydrogenation (mean values ± standard deviations of

enzyme activity measurements at various substrate concentrations, for the data points used see Supplementary Fig. 6c, d). No further oxidation to a carboxylate was observed by enriched S26DH₁.

**Oxidation of 26-OH-DDO to DDO-26-carboxylate.** Enriched S26DH₁ used $K_3[Fe(CN)_6]$ but not $NAD^+$ as electron acceptor for the oxidation of the allylic alcohol **4** to the aldehyde **5**, which is in agreement with its proposed periplasmic location (see Discussion). We tested whether this activity is relevant for the cholesterol catabolic pathway or whether an additional cytoplasmic alcohol dehydrogenase is involved. Using $NAD^+$ as electron acceptor, extracts from *S. denitrificans* grown with cholesterol showed virtually no conversion of 26-OH-DDO **4** to DDO-26-al **5** (<0.1 nmol min⁻¹ mg⁻¹), whereas it served as acceptor for the ready oxidation of DDO-26-al **5** to DDO-26-carboxylate (5 ± 0.2 nmol min⁻¹ mg⁻¹) (mean value ± standard deviation in three replicates). In this assay, formation of an additional product was observed that after ESI-Q-TOF-MS analyses was assigned to 26-OH-DDO (Supplementary Table 1, Fig. 14). Obviously, the NADH formed during DDO-26-al oxidation served as donor for a promiscuous dehydrogenase that reduced the C26 aldehyde to the C26 alcohol.

Recently, a cholesterol induced gene was identified in *S. denitrificans* (WP_154716401) that was hypothesised to encode an aldehyde dehydrogenase (C26-ALDH) involved in cholesterol C26 oxidation[25]. We heterologously expressed this gene in *E. coli*, and extracts from cells producing the recombinant enzyme

indeed catalysed the conversion of DDO-26-al to DDO-26-carboxylate (Supplementary Fig. 15a); in control experiments no such conversions were observed in *E. coli* wild-type without the plasmid expressing WP_154716401. In the presence of MgATP and CoA, the DDO-26-carboxylate formed was further converted to DDO-26-CoA by the previously characterised AMP-forming C26-carboxylate CoA ligase activity[25], thereby initiating β-oxidation of the isoprenoid side chain (Supplementary Fig. 15b).

## Discussion

In this work we identified an enzymatic reaction sequence that allows for the oxidation of an unactivated primary carbon to a carboxylate using water as only hydroxylating agent. In this enzyme cascade the high activation energy barrier for C–H bond hydroxylation of a primary carbon is divided into three individual steps, each of which being energetically and mechanistically plausible under cellular conditions. The C–H bond dissociation energies to carbocations and hydrides relevant for the hydroxylation of tertiary C25 in cholesterol and allylic C26 in desmosterol are 142 kJ mol$^{-1}$ and 117 kJ mol$^{-1}$ lower than that for the primary C26 of cholesterol, respectively (based on gas phase calculations for the isopentane and 2-methyl-2-butene analogues)[35]. Thus, the rationale for the initial formation of a tertiary alcohol during anaerobic cholesterol catabolism[28,29] is to enable the ATP-dependent dehydration to a trisubstituted alkene that is crucial for the subsequent water-dependent hydroxylation of the allylic carbon via a relatively stable allylic carbocation intermediate. The major challenge of oxygen-independent primary carbon activation represents the dehydration of the tertiary alcohol that is achieved by coupling to exergonic ATP hydrolysis (ΔG' ≈ −50 kJ mol$^{-1}$). The phosphate eliminated from the phosphoester intermediate is a much better leaving group than water. A related reaction is known from isoprenoid biosynthesis via the mevalonate pathway: (*R*)-mevalonate-5-diphosphate is ATP-dependently decarboxylated to isopentenyl diphosphate[36,37]. Here, phosphate elimination from an assumed phosphorylated intermediate is accompanied by decarboxylation, which additionally drives the elimination reaction forward (Supplementary Fig. 16). ATP-dependent dehydration has also been reported for the recycling of spontaneously formed NAD(P)H hydrates[38], however it is unclear whether it proceeds via a phosphorylated intermediate.

While a reaction mechanism via tertiary and allylic carbocation intermediates is plausible for the two Mo-dependent hydroxylases S25DH$_1$/S26DH$_1$, it remains uncertain whether this is the case for phosphate elimination from 25-phospho-DDO. If the phosphate elimination would proceed via the identical tertiary carbocation as proposed for S25DH catalysis, the question rises why it is not directly deprotonated by S25DH to the alkene in a single step? Probably, the rebound of the hydroxyl-functionality at the assumed Mo(IV)–OH intermediate to the carbocation is much faster than a competing deprotonation at C24 to an alkene by a putative base.

The reaction cascade identified in this work allows in principle for any enzymatic water-dependent hydroxylation at a primary carbon next to a tertiary one. The genome of *S. denitrificans* contains two further copies of S26DH-like enzymes and they may represent isoenzymes specifically involved in the hydroxylation of allylic methyl groups of intermediates during catabolism of steroids with modified isoprenoid side chains such as β-sitosterol or ergosterol. Hence, a pair of specific Mo-dependent S25DH and S26DH appears to be required for the hydroxylation of primary carbons at the individual isoprenoid chains. The reaction cascade may also be involved in the degradation of non-steroidal isoprenoids or tertiary alcohols. E.g., the anaerobic degradation of

the fuel oxygenate methyl *tert*-butyl ether, an environmental pollutant of global concern, has frequently been described at anoxic environments[39,40]. However, the anoxic degradation of the *tert*-butanol intermediate is unknown but could in principle be accomplished via ATP-dependent dehydration to isobutene, analogous to the described pathway.

Cholesterol degradation in anaerobic bacteria is initiated in the periplasm by AcmA, S25DH$_1$, and probably AcmB dependent conversion into 25-OH-CDO/25-OH-DDO (oxidation to the diene in ring A may also occur later in the pathway). However, subsequent alkene formation is ATP-dependent and therefore has to occur in the cytoplasm. In contrast, subsequent C26 hydroxylation and oxidation to the C26 aldehyde will again take place in the periplasm as evidenced by the N-terminal twin-arginine translocation (TAT) sequence present in S25DH and S26DH (Supplementary Table S3). Thus, initial steps of anaerobic cholesterol degradation involve enzymes alternately accessing their substrates from the periplasm and the cytoplasm (Fig. 5). Though flip-flop of cholesterol between the two leaflets of biological membranes is generally considered fast with rate constants in the $10^4$ s$^{-1}$ range[41–43], it is 5.5-fold faster for 25-OH-cholesterol vs cholesterol[43]. Thus, initial side chain-hydroxylation facilitates the accessibility of cholesterol-derived intermediates from both sides of the cytoplasmic membrane, which appears to be crucial for the reaction cascade involved in water-dependent primary carbon oxidation.

## Methods

**Chemicals and bacterial strains**. The chemicals used were of analytic grade. *Sterolibacterium denitrificans* Chol-1S (DSMZ 13999) and *Thauera aromatica* K172 (DSMZ 6984) were obtained from the Deutsche Sammlung für Mikroorganismen und Zellkulturen (DSMZ, Braunschweig, Germany). *E. coli* BL21 (DE3) and *E. coli* 5α were obtained from New England Biolabs (Frankfurt, Germany).

**Synthesis of non-phosphorylated steroidal compounds**. 25-Hydroxy-cholesta-1,4-dien-3-one (25-OH-CDO) was enzymatically synthesised from the commercially available cholest-4-en-3-one (CDO) using AcmB and S25DH$_1$ from *S. denitrificans* after heterologous production in *E. coli* BL21 (DE3) and *T. aromatica* K172, respectively[30]. The enzyme assays contained 1–2 mM CDO, 50 mM Tris/HCl buffer (pH 7.5), 6% (w/v) 2-hydroxypropyl-β-cyclodextrin (HPCD), 5 mM K$_3$[Fe(CN)$_6$], 2% (v/v) volume cell-free extracts from *E. coli* BL21 producing AcmB, and 4% (v/v) volume cell-free extracts from *T. aromatica* K172 producing S25DH. For synthesis of desmost-1,4-diene-3-one (DDO), soluble proteins from *Sterolibacterium denitrificans* (*S. denitrificans*) were used anaerobically as follows: 1–2 mM 25-OH-CDO, 50 mM Tris/HCl buffer (pH 7.8), 6% (w/v) HPCD, 5 mM ATP, 5 mM MgCl$_2$, and 5% (v/v) volume extracts (cell free) of *S. denitrificans*. 26-OH-DDO and desmost-1,4-diene-3-one-26-al (DDO-26-al) were produced as followed: 1–2 mM DDO, 50 mM Tris/HCl (pH 7.0), 6% (w/v) HPCD, 5 mM K$_3$[Fe(CN)$_6$], and 5% (v/v) volume cell-free extracts of *S. denitrificans*. The enzymatic assays were performed on the 100–1000 mL scale at 30 °C for at least 16 hours. Products were extracted in 2 volumes ethyl acetate, evaporated, and purified by preparative HPLC. For columns and solvents used, see the description of enzyme assays with individual steroids. Samples were lyophilised and solved in 2-propanol before use.

**Synthesis of 25-phospho-cholesta-1,4-dien-3-one**. Bis((9*H*-fluoren-9-yl)methyl) diisopropylphosphoramidite (90%, 29.0 mg, 50.0 µmol, 2.0 eq.) was co-evaporated with dry acetonitrile (2 mL) and dissolved in dry *N*,*N*-dimethylformamide (1 mL). 25-OH-CDO (10 mg, 25 µmol, 1 eq.) and 4,5-dicyanoimidazole (5.9 mg, 50 µmol, 2 eq.) were added and the solution was stirred for 30 min. The solution was cooled to 0 °C and *meta*-chloroperoxybenzoic acid (77%, 6.7 mg, 50 µmol, 2 eq.) was added. After stirring for 15 min, 1,8-diazabicyclo[5.4.0]undec-7-ene (20 µL, 125 µmol, 5 eq.) was added and the mixture was stirred for another 15 min. Subsequently, the solvent was removed under reduced pressure and the crude mixture was purified by preparative HPLC (Solvent A: 10 mM NH$_4$OAc in water; Solvent B: 10 mM NH$_4$OAc in methanol. 0–3 min isocratic 100% A, 3–9 min gradient to 100% B, 9–35 min isocratic 100% B) to obtain the desired phosphate ammonium salt (5 mg, 9.74 µmol, 39%) as a colourless solid. The product was analysed by $^1$H, $^{13}$C, $^{31}$P{$^1$H} NMR, and ESI-HRMS. The spectra are shown in the Supplementary Figs. S2 and S3. $^1$H NMR (400 MHz, methanol-*d₄*): δ = 7.32 (d, *J* = 10.1 Hz, 1H), 6.23 (dd, *J* = 10.1, 2.0 Hz, 1H), 6.08 (t, *J* = 1.7 Hz, 1H), 2.64 – 2.55 (m, 1H), 2.45 – 2.38 (m, 1H), 2.11 (dt, *J* = 13.1, 3.7 Hz, 1H), 2.07 – 1.99 (m, 1H), 1.95 – 1.85 (m, 1H), 1.82 – 1.70 (m, 4H), 1.66 – 1.50 (m, 4H), 1.43 (s, 6H),

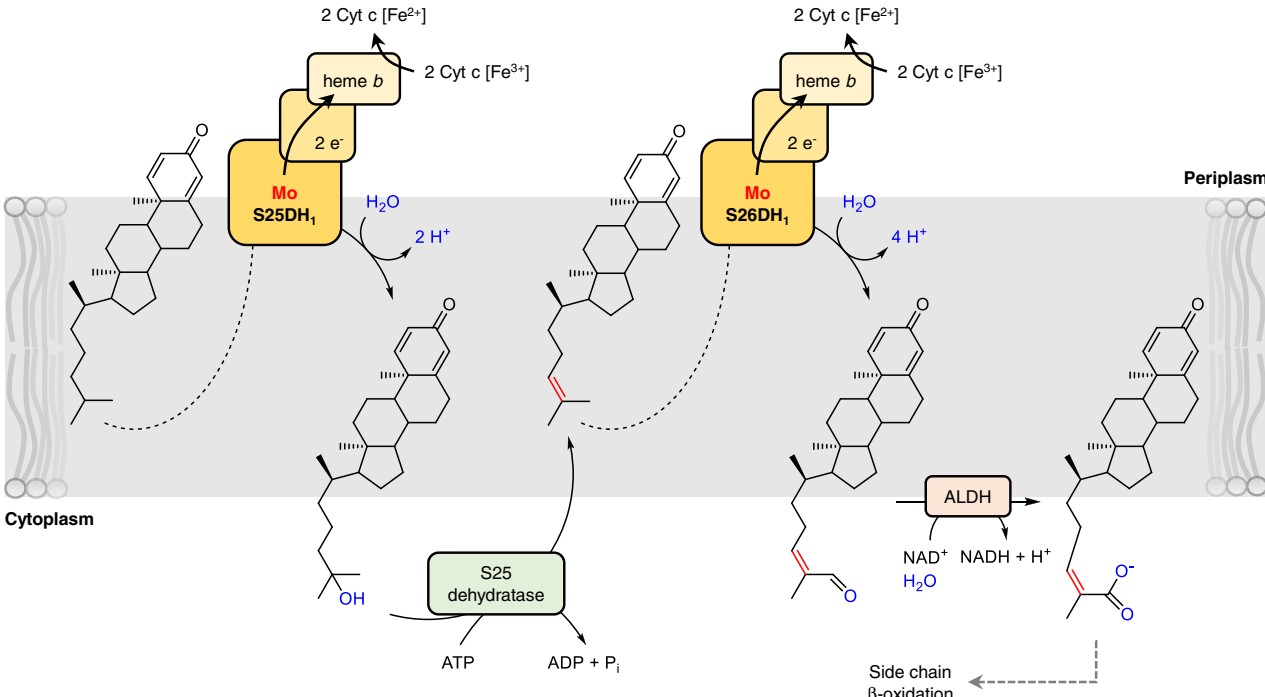

**Fig. 5 Proposed subcellular localisation of initial anaerobic cholesterol degradation steps.** For easier presentation, reduction of Δ24 double bond is omitted. The dotted arrows indicate that hydrophobic side chain needs to be bound by S25DH₁ and S26DH₁. The assignment of S25DH₁ and S26DH₁ to the periplasm are based on their TAT sequence, that of S25 dehydratase and aldehyde dehydrogenase (ALDH) to the cytoplasm on their cytoplasmic co-substrates. Hydroxylations at C25 and C26 abolish polarity of CDO/DDO and improves access from both sides of the cytoplasmic membrane.

1.37 – 1.32 (m, 4H), 1.30 (s, 3H), 1.19 – 1.13 (m, 2H), 1.10 – 1.01 (m, 4H), 0.97 (d, $J = 6.5$ Hz, 3H), 0.82 (s, 3H). ¹³C NMR (101 MHz, methanol-$d_4$): $\delta = 188.8$, 173.9, 159.7, 127.5, 123.9, 81.0 (d, $J = 7.3$ Hz), 57.5, 56.9, 54.3, 45.5, 45.1 (d, $J = 5.4$ Hz), 43.9, 40.9, 37.6, 37.1, 36.8, 35.2, 34.0, 29.2, 28.0 (d, $J = 5.8$ Hz), 28.0 (d, $J = 5.9$ Hz), 25.4, 24.0, 21.8, 19.1, 12.5. ³¹P{¹H} NMR (162 MHz, methanol-$d_4$): $\delta = -3.03$. ESI-HRMS [M−H⁺]⁻: calculated for $C_{27}H_{42}O_5P$ 477.2775, found: 477.2772.

**Culture conditions and preparation of cell extracts.** *S. denitrificans* was cultivated under denitrifying conditions in mineral medium with 2 mM cholesterol and harvested anaerobically in the exponential phase (8000 × *g*, 20 min, and 4 °C) . *T. aromatica* K172 producing S25DH₁ and related enzymes was cultivated under denitrifying conditions using a phosphate-buffered medium with benzoate as carbon source at 30 °C[30]. *E. coli* BL21 was cultivated under aerobic conditions in 2x YT medium (18 g L⁻¹ tryptone, 10 g L⁻¹ yeast extract and 5 g L⁻¹ NaCl) medium at 37 °C. Cells were harvested anaerobically by centrifugation (8000 × *g*, 20 min, and 4 °C) in the late exponential phase. Frozen cells were lysed via a French pressure cell at 137 MPa, using 2 volumes of lysis buffer containing 50 mM MOPS/KOH (pH 7.0) and 0.1 mg DNase I.

**Enzyme assays.** (1) Preliminary tests for the conversion of 25-OH-CDO with cell-free extracts (0.4–0.6 mg ml⁻¹) of *S. denitrificans* were performed in 50 mM MOPS/KOH buffer (pH 7.0), 0.5 mM 25-hydroxy-cholesta-1,4-dien-3-one (25-OH-CDO) or 25-hydroxy-cholest-4-en-3-one (25-OH-CEO), 2 mM electron acceptor ($K_3$[Fe(CN)₆], phenazine methosulfate, 2,6-dichlorophenolindophenol, methylene blue, NADP⁺, or NAD⁺) or 2 mM electron donor (titanium(III) citrate, sodium dithionite, benzyl viologen, methyl viologen, NADPH, or NADH) and 6% (w/v) HPCD at 30 °C under anoxic conditions.

(2) ATP-dependent dehydration of 25-OH-CDO to DDO was determined at 30 °C in the presence of 50 mM Tris/HCl buffer (pH 7.8), 0.2 mM 25-OH-cholesta-1,4-diene-3-one (10–25 mM concentrated in 2-propanol), 5 mM ATP, 5 mM MgCl₂, and 6% (w/v) HPCD. For determination of kinetic parameters and stoichiometry of ATP hydrolysis, soluble proteins from *S. denitrificans* were subjected to (NH₄)₂SO₄ precipitation. At 40% saturation, the activity remained in the supernatant, whereas at 50% the enzyme was found in the precipitate. Enzyme assays were stopped by adding 5 volumes 2-propanol. After twofold-centrifugation, the supernatant was applied to an Acquity H-class UPLC system (Waters) for analysis. An Acquity UPLC CSH C18 (1.7 μm, 2.1 × 100 mm) column was used to separate the steroid compounds using 10 mM aqueous NH₄OAc and an increasing acetonitrile gradient. For the ATP/ADP stoichiometry measurements, the enzymatic assays were stopped using two (v/v) volumes ethyl acetate. The organic

solvent was diluted in 5 (v/v) volumes 2-propanol and centrifuged twice (8,000 × *g*) and loaded onto an Acquity H-class UPLC system. The aqueous phase (20 μL) was acidified using 5 μL 0.25 M HCl and centrifuged twice for 20 min at 4 °C. The supernatant was neutralised using 20 (v/v) volumes MOPS/KOH buffer (pH 7.0). To determine the ATP consumption, an Acquity UPLC HSS T3 (1.8 μM, 2.1 × 100 mm) column was used isocratically with a 10 mM potassium phosphate buffer (pH 6.5)[44].

(3) Conversion of DDO to 26-OH-DDO and DDO-26-al was performed in cell-free extracts (0.4–0.6 mg mL⁻¹) of *S. denitrificans* or enriched S26DH (0.05–0.1 mg mL⁻¹) using 50 mM MOPS/KOH buffer (pH 7.0) in the presence of 0.2 mM DDO (added from 10–25 mM stock in 2-propanol), 5 mM $K_3$[Fe(CN)₆], and 6% (w/v) HPCD at 30 °C under shaking (300 rpm) in an anaerobic glove box (O₂ levels < 2 ppm). Assays were stopped as described above.

(4) Conversion of DDO-26-al to DDO-26-carboxylate was carried out in 50 mM MOPS/KOH buffer (pH 7.0) using 0.2 mM DDO-26-al (10–25 mM stock in 2-propanol), 5 mM NAD⁺, and 6% (w/v) HPCD using cell-free extract (0.4–0.6 mg mL⁻¹) of *S. denitrificans* or cell-free extracts (1.5 mg mL⁻¹) of *E. coli* BL21 producing C26-ALDH at 30 °C under shaking (300 rpm). Assays were stopped as described above.

**Heterologous production of S26DHs in *T. aromatica* K172.** Gene clusters encoding the α₅β₅γ₅-(DH₅), α₆β₆γ₆-(DH₆), α₇β₇γ₇-(DH₇) subunits were amplified using primers suitable for T4 ligation (Supplementary Table S4). The resulting 4.7-kb DNA fragments were HindIII/SpeI double digested; the gene encoding SdhD (δ₄) was amplified using SdhD_for and SdhD_rev primers (Supplementary Table S4), generating a 0.9-kb DNA fragment that was SpeI/XbaI double digested. All fragments were cloned into the broad-host-range vector pIZ1016 (Gmʳ, ori*p*BBR1, Mob⁺, *lacZα, Ptac/lacI*q), and transformed (1.7 kV, 5 ms) into *T. aromatica*. For gene expression, cells were cultivated under denitrifying conditions (30 °C) in phosphate-buffered medium with benzoate, supplemented with 20 μg mL⁻¹ gentamycin and 1 mM IPTG[30].

**Heterologous production of AcmB in *E. coli* BL21.** The gene encoding AcmB was amplified using the primers listed in Supplementary Table S4. The resulting 1.8-kb DNA fragment was SacI/HindIII double digested and cloned into pASK-IBA15plus before transformed into *E. coli* BL21. Induction of AcmB was carried out in 2 YT medium at 20 °C, supplemented with 100 μg mL⁻¹ ampicillin and 20 μg mL⁻¹ anhydrotetracycline. Cells were harvested in the late exponential phase (16 hours induction) and used for further experiments.

**Heterologous production of DDO-26-al dehydrogenase (C26-ALDH) in *E. coli* BL21.** The C26-ALDH gene (WP_154716401) was amplified using the primers listed in Supplementary Table S4. The resulting 1.5-kb DNA Fragment was digested by NheI/NcoI and cloned into pASK-IBA15plus and transformed into *E. coli* BL21. Induction of C26-ALDH was carried out as described for the AcmB gene.

**Enrichment of S26DH₁ after heterologous production in *T. aromatica* K172.** All steps were performed under anaerobic conditions in an anaerobic glove box (95% $N_2$, 5% $H_2$, by vol.; $O_2 < 2$ ppm). The buffers and reagents used were degassed using alternating (20 cycles) $N_2$ (0.5 bar) and vacuum (> -0.9 bar) to reach anaerobicity. Anaerobically harvested cells were lysed with a French pressure cell at 137 MPa using two volumes (w/v) of buffer A (20 mM Tris/HCl pH 7.0, 0.02% [w/v] Tween 20, 0.5 mM dithioerythritol [DTE]). Solubilisation of lyzed cells was carried out for 16 h at 4 °C using 1% (v/v) Tween 20. After ultra-centrifugation at 150,000 × g for 1.5 h, the supernatant was used for enzyme enrichment. The soluble proteins were applied to a DEAE-Sepharose column (75 mL; GE Healthcare) at 7.5 mL min⁻¹ and washed with buffer A. The active protein fraction was eluted by increasing the amount of buffer B (20 mM TRIS/HCl pH 7.0, 0.5 mM DTE, 0.02% [w/v] Tween 20, 500 mM KCl) from 50 mM to 150 mM KCl. Active fractions were concentrated (30-kDa cut-off membrane), diluted in 10 volumes of buffer C (20 mM TRIS/MES pH 6.0, 0.5 mM DTE, 0.01% [w/v] Tween 20), and applied to a Reactive Red 120 column (40 mL; GE Healthcare) at 5 mL min⁻¹. Elution of active fraction was carried out by increasing the gradient of buffer D (20 mM TRIS/HCl pH 8.0, 0.5 mM DTE, 0.01% [w/v] Tween 20) from pH 7 to pH 8. The active protein fraction was concentrated (30-kDa cut-off membrane), and applied to a Superdex 200 column (320 mL) at 2 mL min⁻¹ that was equilibrated with buffer E (20 mM Tris/HCl pH 7.5, 150 mM KCl). The protein eluted in a single peak and was used for further experiments. Uncropped and unprocessed SDS-polyacrylamide gels of enriched enzyme are available in the accompanying Source Data File.

**Spectrophotometric determination of Mo, Fe and *heme* b.** Iron was determined with o-phenantroline using a modified protocol as described[45]. Enriched proteins (5 μL) were added to 245 μL $H_2O$ and acidified with 7.5 μL 25% HCl, mixed, incubated at 80 °C for 10 min, followed by centrifugation at 10,000 × g for 10 min. The supernatant was transferred and 750 μL $H_2O$, 50 μL 10% [w/v] hydroxylamine and 0.1% [w/v] o-phenanthroline were added, mixed and incubated at room temperature for 30 min. The absorbance was measured at 512 nm and compared with a $(NH_4)_2Fe(SO_4)_2$ standard.

Mo was determined colorimetrically with dithiol using a modified protocol[46]. 20 μL proteins were incubated at 60 °C until dryness was reached. The dried product was solved in 250 μL 4 M HCl and heated to 90 °C for 30 min. 100 μL reducing solution (15% [w/v] ascorbic acid and 2% [w/v] citric acid in $H_2O$) were added, mixed and incubated for 5 min at room temperature. 300 μL $H_2O$ and 100 μL dithiol solution (0.1 g zinc dithiol and 0.4 g NaOH diluted in 600 μL ethanol, 31 mL $H_2O$ and 200 μL thioglycollic acid) were added, mixed and incubated for 5 min. Then, 500 μL isoamyl acetate were added and the mixture was shaken vigorously for 1 min. The absorbance was measured in the organic phase at 680 nm and compared with a sodium molybdate dihydrate standard.

The heme *b* content was determined spectrophotometrically at 556 nm after the anaerobically conducted complete reduction of the enzyme with sodium dithionite. The heme-content was calculated using the extinction coefficient of reduced heme *b* ($\varepsilon_{556} = 34.64$ mM⁻¹ cm⁻¹)[47].

**Protein identification by mass spectrometry of peptides.** Proteins were identified by excising the bands of interest from SDS–PAGE gels. After in-gel digestion with trypsin, the resulting peptides were separated by UPLC and identified using a Synapt G2-Si high-resolution mass spectrometry (HRMS) electrospray ionisation quadrupole time-of-flight (ESI-Q-TOF) mass spectrometry system (Waters); the system has been described in detail previously[48].

**LC-ESI-MS analyses of steroid compounds.** Metabolites were analysed by an Acquity I-class UPLC system (Waters) using an Acquity UPLC CSH C18 (1.7 μm, 2.1 × 100 mm) column coupled to a Synapt G2-Si HRMS ESI-Q-TOF device (Waters). For separation, an aqueous 10 mM NH₄OAc/acetonitrile gradient was applied. Samples were measured in positive mode with a capillary voltage of 2 kV, 100 °C source temperature, 450 °C desolvation temperature, 1000 L min⁻¹ $N_2$ desolvation gas flow, and 30 L min⁻¹ $N_2$ cone gas flow. Evaluation of LC-MS metabolites was performed using MassLynx (Waters). Metabolites were identified by their retention times, UV-vis spectra, *m/z* values.

**NMR analyses of steroid compounds.** Deuterated solvents for NMR and reactions were obtained from Armar Chemicals, Switzerland and eurisotop, Germany. NMR spectra were measured on a Bruker Avance Neo 400 MHz or on a Bruker DRX 400 MHz spectrometer. Internal solvent peaks (¹H NMR: methanol-$d_3$: 3.31 ppm; CHCl₃: 7.26 ppm; ¹³C NMR: methanol-$d_4$: 49.0 ppm; CDCl₃: 77.0 ppm)

were used as a reference. Data are reported as follows: chemical shift (δ, ppm), multiplicity (s, singlet; d, doublet; t, triplet; q, quartet; m, multiplet; br. s, broad signal), coupling constant(s) (*J*, Hz), integration. For ³¹P NMR spectra all signals were referenced to an internal standard (PPP).

The chemical structure of cholesta-1,4,24-trien-3-one (DDO, **3**) was deduced from ¹H, ¹³C, and 2D NMR experiments (see Supplementary Fig. S1). In addition to the A-ring signals (¹³C NMR signals at 156.0 (C1), 127.4 (C2), 186.4 (C3), 123.7 (C4), 169.4 (C5) ppm; ¹H NMR signals at 6.98 ppm, 1H, d, *J* = 10.1 Hz, H1; 6.15 ppm, 1H, dd, *J* = 10.1, 1.9 Hz, H2; 5.99 ppm, 1H, 't', *J* = 1.6 Hz, H4), the compound was characterised through ¹³C NMR signals at 125.0 (C24, HSQC cross peak to ¹H NMR δ = 5.01 ppm, 1H, tsept, *J* = 7.1, 1.3 Hz) and 131.0 ppm (C25). The corresponding ¹H NMR signals were identified through HSQC and HMBC pulse experiments. ¹³C NMR (100 MHz, CDCl₃): δ = 186.4, 169.4, 156.0, 131.0, 127.4, 125.0, 123.7, 55.9, 55.4, 52.3, 43.6, 42.6, 39.4, 35.9, 35.5 (2 C), 33.6, 32.9, 28.0, 25.7, 24.6, 24.3, 22.8, 18.6, 18.4, 17.6, 12.0 ppm.

The chemical structure of 26-hydroxycholesta-1,4,24-trien-3-one (26-OH-DDO, **4**) was deduced from ¹H, ¹³C, and 2D NMR experiments (see Supplementary Fig. 5). In addition to the A-ring signals (¹³C NMR signals at 156.0 (C1), 127.4 (C2), 186.4 (C3), 123.7 (C4), 169.5 (C5) ppm; ¹H NMR δ = 7.06 ppm, 1H, d, *J* = 10.1 Hz, H1; 6.22 ppm, 1H, dd, *J* = 10.1, 1.9 Hz, H2; 6.07 ppm, 1H, 't', *J* = 1.6 Hz, H4), the compound was characterised through ¹³C NMR signals at 126.8 ppm (C24, HSQC cross peak to ¹H NMR δ = 5.38 ppm, 1H, t, br, *J* = 7.2 Hz) and 134.3 ppm (C25), as well as ¹³C NMR signals at 69.0 ppm (CH₂OH, C26, HSQC cross peak to ¹H NMR δ = 4.00 ppm, 2H, s) and 13.6 ppm (CH₃, C27, HSQC cross peak to ¹H NMR δ = 1.66 ppm, 3H, s). The corresponding ¹H NMR signals and the *E*-configuration of the C24,C25 double bond were identified through HSQC, HMBC, and 1D NOESY pulse experiments (see Supplementary Fig. 5). ¹³C NMR (100 MHz, CDCl₃): δ = 186.4, 169.5, 156.0, 134.3, 127.4, 126.8, 123.7, 69.0, 55.9, 55.4, 52.3, 43.6, 42.6, 39.4, 35.53, 35.49, 35,45, 33.6, 32.9, 28.1, 24.3, 24.2, 22.8, 18.6, 18.4, 13.6, 12.0 ppm.

The chemical structure of 26-oxocholesta-1,4,24-trien-3-one (DDO-26-al, **5**) was deduced from ¹H, ¹³C, and 2D NMR experiments (see Supplementary Fig. 6). In addition to the A-ring signals (¹³C NMR signals at 156.0 (C1), 127.4 (C2), 186.4 (C3), 123.7 (C4), 169.4 (C5) ppm; ¹H NMR δ = 7.05 ppm, 1H, d, *J* = 10.2 Hz, H1; 6.23 ppm, 1H, dd, *J* = 10.2, 1.9 Hz, H2; 6.07 ppm, 1H, 't', *J* = 1.5 Hz, H4), the compound was characterised through ¹³C NMR signals at 155.3 ppm (C24, HSQC cross peak to ¹H NMR δ = 6.47 ppm, 1H, t, br, *J* = 7.0 Hz) and 139.1 ppm (C25), as well as ¹³C NMR signals at 195.4 ppm (CHO, C26, HSQC cross peak to ¹H NMR δ = 9.39 ppm, 1H, s) and 9.1 ppm (CH₃, C27, HSQC cross peak to ¹H NMR δ = 1.75 ppm, 3H, s). The corresponding ¹H NMR signals and the *E*-configuration of the C24,C25 double bond were identified through HSQC, HMBC, and 1D NOESY pulse experiments (see Supplementary Fig. 6). ¹³C NMR (100 MHz, CDCl₃): δ = 195.4, 186.4, 169.4, 156.0, 155.3, 139.1, 127.4, 123.7, 55.7, 55.4, 52.3, 43.6, 42.7, 39.4, 35.5, 35.4, 34.3, 33.6, 32.8, 28.1, 25.7, 24.3, 22.8, 18.6, 18.3, 12.0, 9.1 ppm.

**Reporting summary.** Further information on research design is available in the Nature Research Reporting Summary linked to this article.

## Data availability
The source data underlying Figs. 2a, b, 3a, 4a, 4b, Supplementary Tables 1 and 2, Supplementary Figs. 4a–d, 7, 8, and 9a, b are provided in the Source Data file. Additional data that support the findings of this study are available from the corresponding author upon reasonable request. Source data are provided with this paper.

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

## Acknowledgements

This work was funded by the German Research Foundation (DFG), RTG 1976, project number 235777276. We thank Georg Fuchs, Freiburg, for continuous scientific advice.

## Author contributions

C.J. performed all enzyme assays, carried UPLC and MS analyses, analysed data, heterologously produced and characterised enzymes, prepared figures and contributed to the writing of the manuscript; S.F. conducted NMR analyses, D.B. and H.J. chemically synthesised and analysed a 25-phospho-desmost-1,4-diene-3-one standard, and contributed to the writing of the paper; M.M. analysed NMR data and contributed to the design of the study; M.B., designed the study, analysed data and wrote the paper. All authors discussed the results and commented on the manuscript.

## Competing interests

The authors declare no competing interests.
