## [Peer Review File · Nature Communications]

REVIEWER COMMENTS

Reviewer #1 (Remarks to the Author):

The manuscript by Jacoby et al. describes the enzymatic activation of a primary carbon in the absence of O₂. The major claims are novel and should be of interest to others in the community and the wider field. More specifically, the described series of reactions initiates the anaerobic catabolism of cholesterol in *Sterolibacterium denitrificans*. However, these reactions will surely represent a paradigm in the anaerobic activation of similar alkanes. For the most part, the experiments are clearly described, the data are clearly presented, and the data are convincing. Some additional characterization of the C26 hydroxylase would strengthen the manuscript.

Specific comments:

1. The manuscript needs to be carefully edited: it contains quite a few grammatical and stylistic errors. The following are cited as examples (the list is not exhaustive): "non-activated" should be "un-activated" (title, elsewhere); "no option" should be "not an option" (l. 49); "proceed" should be "proceeds" (l. 54); "During" should be "In" (l. 62); insert a comma at the end of line 83; "an artificial electron acceptor" (1.88); "analogues of it" should be "its analogues" (1. 97); "a so far unknown" should be "previously undescribed" (l. 104); "during" should be "using" (1. 125); "anaerobically conducted assays" should be "assays performed anaerobically" (1. 171); "putatively" should be "potentially" (l. 210); and "can hardly" should be "is unlikely to" (l. 223).
2. The biologically relevant form of the reaction product is a carboxylate, not a carboxylic acid. It should be depicted and referred to as such throughout the manuscript (Figs. 1 and, l. 27, 1. 107, and elsewhere).
3. The statement that "isoprenoid side chain is cleaved ... prior to steran skeleton degradation" is incorrect (l. 63). Capyk et al. (*J Biol Chem* 286, 40717) provided evidence that these proceed concurrently to at least some extent.
4. The first two sentences of the Results repeat the Introduction. The section should begin "To elucidate the enzymology responsible for production of CDO-26-oate" (l. 115).
5. In Figure 4a, "DH5" should be relabeled as "S26DH1" (i.e., the identity of this enzyme needs to be much clearer).
6. The authors should determine the Mo, Fe, S, and heme content of their preparation of S26DH1. As it stands, the prosthetic groups of the enzyme are solely inferred from bioinformatics.
7. The conditions used to achieve anaerobicity for the enzyme purification and characterization (e.g., kinetics) need to be described. This should include enough detail so that the reader is not obliged to look up additional papers. In addition, the authors should provide estimates of O₂ levels in these manipulations.

Reviewer #2 (Remarks to the Author):

Boll and colleagues have discovered a sequential enzymatic process that enables hydroxylation of inert primary C-H bonds with water, which involves two molybdenum-dependent hydroxylases and one ATP-dependent dehydratase. This biological process is new and therefore this work represents interests and new insights for enzymologists and organic chemists. The authors performed thorough analytics and therefore the results and mechanisms in the manuscript are convincing. After evaluating the manuscript carefully, I think the manuscript needs some improvements at this stage before it can be accepted in top-tier Nature Communications. Concerns are shown below. I hope that the authors will find the comments helpful to improve the manuscript.

- 1) In the introduction, the oxygen-independent enzymatic oxidation of C-H bonds is kind of limited. There are other enzymes that can perform similar reactions without the use of O₂, e.g. peroxygenases, peroxidases, etc. It would be great if the authors can broaden the interest in the introduction.

2) I suggest that in the supplementary NMR spectra the molecule structure should be included, as well as the assignment of the peaks, this is will be more straightforward to catch the information.

3) The authors indicated that "the reaction cascade identified in this work allows in principle for any enzymatic water-dependent hydroxylation at a primary carbon next to a tertiary one." Did the authors extend the substrate scope to see the specificity/genericity of the present enzymatic cascade? For example, what about the hydroxylation and oxidation of cholecalciferol and other similar compounds?

4) The labelling of the compounds 1,2,3... are confusing, the compound number were sometimes not in bold, it would be great if the authors could label the compound more consistently and precisely like the way usually seen in organic synthesis.

5) Is there any specific reason that the experiments were performed with crude extracts? In this way key information on the catalytic capability of the enzymes (e.g. TTN, TOF) are missing.

6) The typo errors need to be checked: e.g. line 165, "my" should read "by"; in Figure 1 "26-OH-CDO" was selected and the box was still there, etc.

Reviewer #3 (Remarks to the Author):

Specific Comments and Suggestions for Revision:

The current study is the latest in a series of studies by M. Boll and co-workers aimed at elucidating the enzymatic reactions involved in steroid catabolism in the absence of oxygen by anaerobic bacteria. A new class of molybdenum-dependent steroid dehydrogenase enzymes capable of oxygen-independent hydroxylation of the C-25 group of cholesterol was first reported by Chiang et al., in 2008. In a recent paper by C. Jacoby et al. (mBio 9(3), 2018), four molybdenum-dependent dehydrogenase isozymes were overexpressed in bacteria, isolated and characterized.

In my opinion, the study provides important new information regarding oxygen-independent hydroxylation reactions catalyzed by bacterial enzymes. A large amount of relevant data is presented in the supplementary section. This is a novel and interesting study. However I found some ambiguity in the characterization of the final two steps of the reaction sequence (as outlined below). The manuscript is well written. There are a few awkwardly worded phrases and unnecessary sentences. Moreover, I found the use of semi-colons to be excessive. I think that the use of periods and shorter sentences would improve readability.

Pg. 2 (Abstract), line 28, and elsewhere in the manuscript – I suggest deleting the word, 'unprecedented', as this descriptor is un-necessary.

Pg. 3 (Introduction), lines 39-47 — In my opinion, the Introduction section is too long. It seems to me that the first paragraph of the Introduction is un-necessary and could be deleted in order to simplify and shorten the introduction.

Pg. 3 (Introduction), line 49 — I suggest replacing the words, 'no option', with, 'not an option'.

Pg. 3 (Introduction), line 54 — The word, 'proceed', should be written as, 'proceeds'.

Pg. 3 (Introduction), line 62 — The phrase, '... globally important bacterial degradation of cholesterol ...', should be justified. How and why is bacterial degradation of cholesterol important on a global scale?

Pg. 4 (Introduction), line 65 — I suggest the phrase, '... during persistence in macrophages ...', in this sentence is confusing and un-necessary and should be deleted in order to simplify the sentence.

Pg. 4 (Introduction), line 67 — I suggest adding replacing the abbreviation, 'Cyp450s', with, 'cytochrome P450 enzymes'. It should be noted that the widely used abbreviation for cytochrome P450 is CYP, not, cypP450.

Pg. 4 (Figure 1 legend), line 80 — It would be helpful if the abbreviation CDO, which is defined in the figure, was also defined in the figure legend.

Pg. 5 (Introduction), lines 98 and 101 — The words, 'rationale' and 'chaperone' are misspelled here and elsewhere in the manuscript.

Pg. 5 (Introduction), line 105 — I suggest simplifying the phrase, '... primary one would involve a so far unknown enzymology.' by re-writing it as, '... primary alcohol involves unknown enzymology'.

Pg. 7 (Results), lines 111-121 — In my opinion, this paragraph does not belong in the Results section. It is perhaps more suitable for the Methods section.

Pg. 9 (Results), lines 165-166 — It would be helpful if the units (eg. 1.2 ± 0.1 ATP) were specified.

Pg. 9 (Results), line 170 — It would be helpful if the authors specified what was meant by 'the latter'. What are they specifically referring to?

Pg. 10 (Results), line 185 — I suggest re-writing the phrase, '... in cis to the C24 ...', as, '... in the cis position relative to the C24 ...'.

Pg. 10 (Results), lines 188-190 — It seems to me that the sentence stating that the oxidation of the C-26 alcohol to a C-26 aldehyde needs some explanation because the authors state (on line 172 and in Fig. 3) that cell-free extracts of *Sterolibacterium denitrificans* produced two products, 26-OH-DDO and DDO-26-al. It is unclear to me why the authors then state that 26-OH-DDO was further oxidized to DDO-26-al. Are the authors suggesting that only the alcohol product was formed in the initial reaction and that the alcohol product is converted to the aldehyde product in a subsequent reaction? Is this oxidation reaction catalyzed by a new enzyme such as alcohol dehydrogenase or are both reactions catalyzed by S26DH1? To my mind, this sub-section is ambiguous.

Pg. 11 (Results), line 54 — I suggest inserting the word, 'enzymes', between 'DH5-7' and 'may'.

Pg. 12 (Results), lines 214-215 — In my opinion, this sentence is worded awkwardly and should be re-written.

Pg. 12 (Results), line 222 — It is unclear to me how ions obtained by TOF-MS identified the bands as the subunits of S26DH1. Do the authors mean that the masses of the ions correspond to the masses of the subunits? It would be helpful if the authors clearly explained the method of identification.

Pg. 12 (Results), lines 228-235 — As noted above, the authors state (on line 172 and in Fig. 3) that a cell-free extract produced two products, 26-OH-DDO and DDO-26-al, yet the authors now imply that S26DH1 first produced 26-OH-DDO and then catalyzed its dehydrogenation to DDO-26-al. The dehydrogenation of an alcohol to an aldehyde is typically catalyzed by an alcohol dehydrogenase. Are the authors now suggesting that S26DH1 catalyzes separately a hydroxylation reaction and an alcohol dehydrogenase reaction? Is there evidence that S25DH1-4, which are related enzymes, can catalyze both a hydroxylation reaction and an alcohol dehydrogenase reaction? I think a more complete explanation is needed.

Pg. 13 (Results), lines 236-258 — I found this section of the Results to be confusing. If I understand the section correctly, the authors are suggesting the involvement of two enzymes, an alcohol dehydrogenase and an aldehyde dehydrogenase. The alcohol dehydrogenase catalyzes the dehydrogenation of the C-26 alcohol to an aldehyde, which seems to contradict the statement on the previous page regarding S26DH1, while the aldehyde dehydrogenase catalyzes the dehydrogenation of the C-26 aldehyde to a carboxylic acid. Have the authors isolated and identified this new aldehyde dehydrogenase from *E. coli*? Again, I think a fuller explanation is needed.

Pg. 15 (Discussion), line 287 — It is not clear to me what, 'it', (i.e. If it would proceed ...) refers to. It would be helpful if the authors specified what was meant by 'it'. What are they specifically referring to?

Point to point response (own statements in *blue italics*)

Reviewer 1:

1. All grammatical and stylistic errors were edited as suggested by the reviewer:

“Non-acitvated” was replaced by “ unactivated”; “no option was replaced by “not an option”; “proceed” was replaced by “proceeds”; “During” was replaced by “In”; “During” was replaced by “In”; a comma was inserted at the end of line 83; “analogues of it” was replaced by “its analogues”; “a so far unknown” was replaced by “previously undescribed”; “during” was replaced by “using”; “anaerobically conducted assays” was replaced by “assays performed anaerobically”; “putatively” was replaced by “potentially”; “can hardly” was replaced by “is unlikely to”.

2. *‘Carboxylic acid’ is now referred to as ‘carboxylate’ throughout the manuscript.*

3. The statement that “isoprenoid side chain is cleaved ... prior to stearn skeleton degradation” is incorrect (l. 63). Capyk et al. (J Biol Chem 286, 40717) provided evidence that these proceed concurrently to at least some extent.

We fully agree with the reviewer, but also realized that the statement about the order of side-chain/steran skeleton degradation is not really relevant for the presented study, and we have omitted this statement.

4. *The title and the first two sentences of the first paragraph of the Results section were deleted according to the reviewers suggestion.*

5. In Figure 4a, “DH5” should be relabelled as “S26DH1” (i.e., the identity of this enzyme needs to much clearer)

We re-labelled the enzyme with S26DH1 and added in the legend a statement, that this enzyme was previously referred to as DH5: ‘The previously assigned DH₅ is now referred to as S26DH₁ (marked in red).’

6. The authors should determine the Mo, Fe, S, and heme content of their preparation of S26DH1. As it stands, the prosthetic groups of the enzyme are solely inferred from bioinformatics.

We determined the Mo, non-heme and heme iron by colorimetric techniques. The numbers determined are in very good agreement with numbers deduced from bioinformatics and were 0.81 Mo, 19.7 non-heme Fe, and 0.89 heme b (Expected were 1:19:1). We added the following statement in the results section and described and referenced the procedures applied in the materials and methods part:

‘Amino acid sequence similarities of S26DH₁ with characterized S25DHs²⁸ suggested the presence of one Mo-cofactor, four [4Fe-4S] clusters, one [3Fe-4S] cluster, and a heme b. The metal content of S26DH1 determined by colorimetric procedures was 0.81 ± 0.03 mol Mo, 19.7 ± 0.4 mol Fe and 0.89 ± 0.01 mol heme b per mol S26DH₁, respectively. These values are in good agreement with the expected cofactor content and with the values determined for other S25DHs²⁸ .’

7. The conditions used to achieve anaerobicity for the enzyme purification and characterization (e.g., kinetics) need to be described. This should include enough detail so that the reader is not obliged to look up additional papers. In addition, the authors should provide estimates of O₂ levels in these manipulations.

For clarification, we added the following statement in the materials and methods section, paragraph 'Enrichment of S26DH1)...':

'All steps were performed under anaerobic conditions in an anaerobic glove box (95% N₂, 5% H₂, by vol.; O₂ < 2 ppm). The buffers and reagents used were degassed using alternating (20 cycles) N₂ (0.5 bar) and vacuum (> -0.9 bar) to reach anaerobicity.'

Reviewer 2:

1) In the introduction, the oxygen-independent enzymatic oxidation of C-H bonds is kind of limited. There are other enzymes that can perform similar reactions without the use of O₂, e.g. peroxygenases, peroxidases, etc. It would be great if the authors can broaden the interest in the introduction.

We agree with the reviewer that indeed peroxygenase or peroxidases may be involved in similar reactions. However, we have the dilemma that reviewer #3 already believes that the first paragraph is too long or even dispensable. As a kind of a compromise, we deleted the long second sentence of the first paragraph, and added the statement that peroxidases or peroxygenases may be involved in C-H bond activations. We hope that the reviewer agrees with this compromise.

2) I suggest that in the supplementary NMR spectra the molecule structure should be included, as well as the assignment of the peaks, this is will be more straightforward to catch the information.

The molecule structure is now included. However, it is really not standard to assign NMR peaks to the structure. This would result in a highly messy figure where nothing could really be followed anymore. Notably, the ppm values determined for the carbon atoms are listed in the Materials and Methods section, so any reader with basic knowledge of NMR has the possibility to assign peaks to the structure.

3) The authors indicate that "the reaction cascade identified in this work allows in principle for any enzymatic water-dependent hydroxylation at a primary carbon next to a tertiary one." Did the authors extend the substrate scope to see the specificity/genericity of the present enzymatic cascade? For example, what about the hydroxylation and oxidation of cholecalciferol and other similar compound?

In very preliminary assays, we tested the only available 25-OH-sitosta-1,4-diene-3-one (converted to the corresponding dehydrated product) and 25-OH-VitD3 (not converted) as potential alternative substrates for the kinase/dehydratase activity in extracts of cells grown with cholesterol. We believe that extensive substrate range studies will be on the agenda once the kinase/dehydratase has been identified and isolated. The reviewer may also take into account that different kinase/dehydratases are involved in the conversion of different tertiary alcohol substrates, and we can only determine in extracts the one that is induced during growth with cholesterol. Probably others are induced under different growth conditions, similar as observed for the four isoenzymes of S25DH.

4) The labelling of the compounds 1,2,3... are confusing, the compound number were sometimes not in bold, it would be great if the authors could label compound more consistently and precisely like the way usually seen in organic synthesis.

This has been changed according to the reviewer's suggestion.

5) Is there any specific reason that the experiments were performed with crude extracts? In this way key information on the catalytic capability of the enzymes (e.g. TTN, TOF) are missing.

The ATP-dependent phosphorylation and dehydration was catalysed by cell-free extracts, the enzyme catalysing this reactions is not known and needs to be isolated in the future. So we can't give TTN, TOF for this reaction. Instead, we have given K_m and V_{max} for all enzymatic reactions. The hydration of the alkene was also studied with purified S26DH1 that has been isolated and studied in this work. Again, we have given the basic K_m and V_{max} values for the reaction catalysed (Michaelis-Menten curves, see SI). The deduced catalytic number using the calculated mass from the subunits would be around $0.5\ s^{-1}$, which adds no information on the initial scope of this study. We really prefer to conduct detailed kinetic assays rather with all purified enzymes, individually.

6) The typo errors need to be checked: E.g. line 165, "my" should read "by", in Figure 1 "26-OH-CDO" was selected and the box was still there, etc.

corrected

Reviewer 3:

Pg. 2 (Abstract), line 28, and elsewhere in the manuscript – I suggest deleting the word, 'unprecedented', as this descriptor is un-necessary.

The word "unprecedented" was deleted throughout the manuscript.

Pg. 3 (Introduction), lines 39-47 – In my opinion. The introduction section is too long. It seems to me that the first paragraph of the Introduction is un-necessary and could be deleted in order to simplify and shorten the introduction.

We have deleted the long second sentence of the first paragraph. However, reviewer #2 urged us even to add more information about peroxygenases/peroxidases involved in C-H-bond activations (which is now briefly added). So we believe that we have now found a compromise to satisfy both reviewers.

Pg. 3 (Introduction), line 49 – I suggest replacing the words, 'no option', with 'not an option'.

"no option" was replaced by "not an option" throughout the manuscript.

Pg. 3 (Introduction), line 54 – The word, 'proceed', should be written as, 'proceeds'.

Corrected

Pg. 3 (Introduction), line 62 – The phrase, '... globally important bacterial degradation of cholesterol...', should be justified. How and why is bacterial degradation of cholesterol important on a global scale?

We added now a corresponding justification sentence which was extended to steroids in general and not restricted to cholesterol:

'Bacterial degradation of steroids is of global importance for biomass decomposition, removal of environmental pollutants and for intracellular survival of Mycobacterium tuberculosis and other pathogens'

Pg. 4 (Introduction), line 65 – I suggest the phrase, ‘... during persistence in macrophages ...’, in this sentence is confusing and un-necessary and should be deleted in order to simplify the sentence.

The phrase “... during persistence in macrophages...” was deleted.

Pg. 4 (Introduction), line 67 – I suggest adding replacing the abbreviation. ‘Cyp450s’, with, ‘cytochrome P450 enzymes’. It should be noted that the widely used abbreviation for cytochrome P450 is CYP, not, cypP450.

“Cyp450” was replaced by “cytochrome P450 monooxygenase” throughout the manuscript.

Pg. 4 (Fugure 1 legend), line 80 – It would be helpful if the abbreviation CDO, which is defined in the figure, was also defined in the figure legend.

The following abbreviations were included in the figure legend. 25-OH-CDO (25-OH-cholest-1,4-diene-3-one); CDO-26-carboxylate (cholest-1,4-diene-3-one-26-carboxylate); 26-OH-CDO (26-OH-cholest-1,4-diene-3-one).

Pg. 5 (Introduction), lines 98 and 101 – The words, ‘rationale’ and chaperone’ are mis-spelled here and elsewhere in the manuscript.

Corrected.

Pg. 5 (Introduction), line 105 – I suggest simplifying the phrase, ‘... primary one would involve a so far unknown enzymology.’ By re-writing it as, ‘... primary alcohol involves unknown enzymology’ by re-writing it as, ‘... primary alcohol involves unknown enzymology’.

Thank you for this suggestion: The phrase ‘... primary one would involve a so far unknown enzymology’. Was rephrased to “... primary alcohol involves a previously undescribed enzymology”.

Pg. 7 (Results), lines 111-121 – In my opinion, this paragraph does not belong in the Results section. It is perhaps more suitable for the Methods section.

The first paragraph of the Results section was shortened and integrated into the previously second paragraph. According to reviewer #1, the first two sentences and the title have already been deleted from this paragraph. However, the rest of the paragraph indeed presents results, though negative. We believe it is important to state that the assays for demonstrating a potential oxidation of the tertiary alcohol somehow to the carboxylate have been carried out. Only their negative outcome motivated us to search for alternative solutions.

Pg. 9 (Results), lines 165-166 – It would be helpful if the units (eg. 1.2 ± 0.1 ATP) were specified.

The phrase was modified as followed: “UPLC analyses of adenosine nucleotides revealed that 1.2 ± 0.1 mol ATP were hydrolysed to 0.9 ± 0.1 mol ADP and 0.3 ± 0.1 mol AMP per mol 25-OH-CDO consumed.

Pg. 9 (Results), line 170 – It would be helpful if the authors specified what was meant by ‘the latter’. What are they specifically referring to?

The latter refers to DDO. “the latter” was replaced by “DDO”.

Pg. 10 (Results), line 185 – I suggest re-writing the phrase, ‘... in cis to the C24 ...’, as, ‘... in the position relative to the C24...’.

Rephrased.

Pg. 10 (Results), line 188-190 – It seems to me that the sentence stating that the oxidation of the C-26 alcohol to a C-26 aldehyde needs some explanation because the authors state (on line 172 and in Fig. 3) that cell-free extracts of *Sterolibacterium denitrificans* produced two products, 26-OH-DDO and DDO-26-al. It is unclear to me why the authors then state that 26-OH-DDO was further oxidized to DDO-26-al. Are the authors suggesting that only the alcohol product was formed in the initial reaction and that the alcohol product is converted to the aldehyde product in a subsequent reaction? Is this oxidation reaction catalysed by a new enzyme such as alcohol dehydrogenase or are both reactions catalysed by S26DH1? To my mind, this sub-section is ambiguous

The time course shown in Fig. 3a (right panel) clearly shows that first product 4 and then product 5 is formed leaving little doubt about the order of product formation (first 4 and then 5). Further, there is no other explanation of the product pattern as: (i) hydroxylation of the alkene, followed by the dehydrogenation of the alcohol formed (hydration first, followed by dehydrogenation). This is exactly the identical order of reaction we observed with heterologously expressed S25DH1 (see below).

To exclude any misunderstanding we rephrased this section (e.g., stating ...converted DDO to product 4 that was subsequently converted to 5). Further, we omitted the proposed involvement of the Mo-factor in Fig. 3b, and simply indicate that 3 is converted to 4 via the proposed allylic intermediate, that is then dehydrogenated

Pg. 11 (Results), line 54 — I suggest inserting the word, ‘enzymes’, between ‘DH5-7’ and ‘may’.

The word “enzymes” was inserted.

Pg. 12 (Results), lines 214-215 — In my opinion, this sentence is worded awkwardly and should be re-written.

*This sentence was rephrased: ‘Heterologously produced S26DH₁ was enriched from *T. aromatica* extracts by three chromatographic steps under anaerobic conditions using a modified protocol established for SDH₁₋₄²⁸*

Pg. 12 (Results), line 222 — It is unclear to me how ions obtained by TOF-MS identified the bands as the subunits of S26DH1. Do the authors mean that the masses of the ions correspond to the masses of the subunits? It would be helpful if the authors clearly explained the method of identification.

*This part was rephrased: ‘The masses of the ions obtained from tryptic peptides confirmed the presence of the α,β,γ -subunits of S26DH₁ from *S. denitrificans*, those obtained from the band eluting at 43 kDa were assigned to a co-purified....’*

Pg. 12 (Results), lines 228-235 — As noted above, the authors state (on line 172 and in Fig. 3) that a cell-free extract produced two products, 26-OH-DDO and DDO-26-al, yet the authors now imply that S26DH1 first produced 26-OH-DDO and then catalyzed its dehydrogenation to DDO-26-al. The dehydrogenation of an alcohol to an aldehyde is typically catalyzed by an alcohol dehydrogenase. Are the authors now suggesting that S26DH1 catalyzes separately a hydroxylation reaction and an alcohol dehydrogenase reaction? Is there evidence that S25DH1-4, which are related enzymes, can catalyze both a hydroxylation reaction and an alcohol dehydrogenase reaction? I think a more complete explanation is needed.

Yes, it is exactly as stated by the reviewer! S26DH1 catalyses both: hydroxylation of S26 and then the dehydrogenation of the primary alcohol to an aldehyde. Such a reaction cannot be catalysed by S25DH1-4, because they form a tertiary alcohol at C25 that cannot be further oxidized. We believe

that the story should be clear as: (i) we have indicated and shown that heterologously produced S26DH1 catalysed both, hydroxylation and dehydrogenation, whereas T. aromatica extracts where the enzyme has been heterologously produced did not catalyze such a reaction, (ii) we have indicated that only ferricyanide but not NAD⁺, the usual acceptor for alcohol dehydrogenation, serves as acceptor for the dehydrogenation of the alcohol to the aldehyde. We really cannot see any alternative to our conclusion from the data that S26DH1 catalyzes both, hydroxylation and dehydrogenation. This makes sense because oxidation of the alcohol with NAD⁺ is less favourable than with cytochrome c (the assumed periplasmic acceptor of S26DH1).

Pg. 13 (Results), lines 236-258 — I found this section of the Results to be confusing. If I understand the section correctly, the authors are suggesting the involvement of two enzymes, an alcohol dehydrogenase and an aldehyde dehydrogenase. The alcohol dehydrogenase catalyzes the dehydrogenation of the C-26 alcohol to an aldehyde, which seems to contradict the statement on the previous page regarding S26DH1, while the aldehyde dehydrogenase catalyzes the dehydrogenation of the C-26 aldehyde to a carboxylic acid. Have the authors isolated and identified this new aldehyde dehydrogenase from E. coli? Again, I think a fuller explanation is needed.

We really tried hard several times but we still cannot follow the problems/obvious contradictions of the reviewer concerning this point: we have clearly stated that S26DH1 catalyses BOTH hydroxylation of C26 and the subsequent oxidation of the alcohol to aldehyde. Both reactions must take place in the periplasm due to the periplasmic location of S26DH1, with cytochrome c serving as electron acceptor and not NAD⁺.

In the last paragraph of the results section, we have explicitly stated and shown that heterologously produced aldehyde dehydrogenase (WP_154716401) catalyzes the oxidation of the aldehyde to the carboxylic acid. Of course not the enzyme from E. coli (there is no such enzyme in E. coli), but we heterologously produced the enzyme from S. denitrificans in E. coli.

Pg. 15 (Discussion), line 287 — It is not clear to me what, 'it', (i.e. If it would proceed ...) refers to. It would be helpful if the authors specified what was meant by 'it'. What are they specifically referring to?

"it" was replaced by "the phosphate elimination".

REVIEWERS' COMMENTS:

Reviewer #1 (Remarks to the Author):

The authors have thoroughly addressed all of my comments.

Reviewer #2 (Remarks to the Author):

The authors have spent some efforts in addressing three reviewers' comments on the manuscript. Personally I think that based on the novelty of the work, the manuscript can be accepted with very few minor effort.

For the introduction, I still believe that the authors need to re-organize the introduction on C-H bonds hydroxylation. Please note that the quality of an introduction is not necessarily bound to its length. Please add references to the peroxidases and peroxygenases.

Reviewer #3 (Remarks to the Author):

I reviewed the original submission by C. Jacoby and co-workers. In the revised manuscript, the authors addressed most of my initial concerns and those of the other reviewers. Grammar, wording and sentence structure have been improved. Some sentences were re-written for the revised manuscript and some new phrases were added. I have no further concerns.

Response to reviewers comments (blue italics)

Reviewer #2 (Remarks to the Author):

The authors have spent some efforts in addressing three reviewers' comments on the manuscript. Personally I think that based on the novelty of the work, the manuscript can be accepted with very few minor effort.

For the introduction, I still believe that the authors need to re-organize the introduction on C-H bonds hydroxylation. Please note that the quality of an introduction is not necessarily bound to its length. Please add references to the peroxidases and peroxygenases.

We added three references and two additional sentences for emphasizing the importance of enzymatic C-H-bond activations.